# The Nrf2 in Obesity: A Friend or Foe?

**DOI:** 10.3390/antiox11102067

**Published:** 2022-10-20

**Authors:** Yudong Xia, Xiaoying Zhai, Yanning Qiu, Xuemei Lu, Yi Jiao

**Affiliations:** State Key Laboratory of Pathogenesis, Prevention and Treatment of High Incidence Diseases in Central Asia, Xinjiang Key Laboratory of Molecular Biology for Endemic Diseases, Department of Biochemistry and Molecular Biology, School of Basic Medical Sciences, Xinjiang Medical University, Urumqi 830011, China

**Keywords:** Nrf2, obesity, lipid metabolism, chronic inflammation, oxidative stress

## Abstract

Obesity and its complications have become serious global health concerns recently and increasing work has been carried out to explicate the underlying mechanism of the disease development. The recognized correlations suggest oxidative stress and inflammation in expanding adipose tissue with excessive fat accumulation play important roles in the pathogenesis of obesity, as well as its associated metabolic syndromes. In adipose tissue, obesity-mediated insulin resistance strongly correlates with increased oxidative stress and inflammation. Nuclear factor erythroid 2-related factor 2 (Nrf2) has been described as a key modulator of antioxidant signaling, which regulates the transcription of various genes coding antioxidant enzymes and cytoprotective proteins. Furthermore, an increasing number of studies have demonstrated that Nrf2 is a pivotal target of obesity and its related metabolic disorders. However, its effects are controversial and even contradictory. This review aims to clarify the complicated interplay among Nrf2, oxidative stress, lipid metabolism, insulin signaling and chronic inflammation in obesity. Elucidating the implications of Nrf2 modulation on obesity would provide novel insights for potential therapeutic approaches in obesity and its comorbidities.

## 1. Introduction

Obesity, characterized by excess adiposity, has become a major public health problem worldwide which adversely affects mental and physical health and imposes a substantial socio-economic burden [1,2]. The prevalence of obesity has doubled in more than 70 countries around the world and has constantly increased since 1980 [3]. Moreover, health issues resulting from overweight and/or obese have affected more than 2 billion people [4]. Obesity increases the risk for an extensive range of diseases, including type 2 diabetes mellitus (T2DM) [5], ischemic heart disease [6,7], hypertension [8] and certain types of cancer [9]. Obesity has been positively correlated with higher all-cause mortality risk [10,11]. What is more, the evidence cooperatively indicates that obesity accelerates ageing [12].

Although a lot of research has focused on obesity and its comorbidities, the underlying mechanisms have not yet been made fully clear. In recent years, the crucial role of obesity in the development of T2DM has become progressively fair due to their association with insulin resistance (IR). The oblique common cause of IR is obesity, while IR is the pathogenic basis of obesity and T2DM [13,14,15]. Interestingly, obesity-induced IR is strongly associated with chronic low-grade inflammation in white adipose tissue (WAT). Both in vitro and in vivo studies have suggested a relationship between WAT inflammation and the progression of T2DM [16,17,18]. However, the mechanism linking obesity and WAT inflammation with IR has not been entirely explicated.

Growing studies have shown that oxidative stress, which is defined by excessive endogenous reactive oxygen species (ROS) levels [19,20], plays a critical role in the pathogenesis and development of obesity and T2DM. An extended hyperglycemic condition leads to a periodic and sustained increase in ROS levels [21,22], and elevates the levels of free fatty acids (FFA) released by WAT of obese individuals, promoting ROS production [22,23]. In turn, the hyperglycemia-mediated ROS overproduction and the increased ROS generation within obese WAT leads to the secretion of metabolically adverse adipocytokine patterns, and provokes inflammatory responses, contributing to IR through the interaction between inflammatory molecules (certain adipocytokines) and insulin signaling [16,17,24]. Collectively, they may serve both as the cause and result of each other, thus forming a vicious circle.

Maintaining redox balance depends on a powerful antioxidant system that counteracts oxidative stress. The nuclear factor erythroid 2-related factor 2 (Nrf2), belonging to the family of the cap n’ collar transcription factors, is a key regulator of the cellular response to oxidative stress by controlling the expression of antioxidant and detoxification enzymes to eliminate excess ROS [25,26,27]. Nrf2 is constitutively expressed under homeostatic conditions, localized in the cytoplasm, which binds to the Kelch-like ECH-associated protein 1 (Keap1), facilitating the ubiquitination and proteasomal degradation [28]. Under oxidative stress conditions, certain reactive cysteines in Keap1 are modified by ROS, and then Nrf2 dissociates from the Keap1 and translocates to the nucleus [29,30,31]. Subsequently, Nrf2 binds to antioxidant response element (ARE; GTGACNNNGC) present in the regulatory region of the Nrf2 target genes that codes antioxidant and detoxification proteins, thereby inducing the gene transcription [32,33] (Figure 1).

Nrf2 is like a coin with two sides. Although Nrf2 activation is an important antioxidant modulator in normal cells, this effect is also one of the main factors involved in the enhanced resistance to chemotherapy and radiotherapy in cancer cells since many studies are focused to reduce Nrf2 activation in chemoresistant and radioresistant cancer cells [34,35,36]. More remarkably, Nrf2 activation by antidiabetic agents has been reported to accelerate tumor metastasis [37].

It has certainly been acknowledged that Nrf2 is indispensable for the regulation of inflammatory responses [18,38,39]. Given that obesity has been closely associated with inflammation and oxidative stress, the potential protective function of Nrf2 is of great interest. However, the effects of Nrf2 are controversial and even contradictory. Table 1 summarizes the recent findings presented on the different roles of Nrf2 signaling in animal models of diet-induced obesity (DIO).

## 2. Is Nrf2 Up-Regulated or Down-Regulated in Response to Obesity? Is It an Antagonist or a Protector?

Although many studies have established that obesity can increase oxidative stress [22,23,24,48,60,61,62], as a master regulator of antioxidant defense, Nrf2 appears to display differential expression patterns in obesity. In a recent study, the WAT of wild-type (WT) mice and ob/ob mice fed high-fat diet (HFD; 41% kcal fat) and normal diet (ND; 11% kcal fat), respectively, for 16 weeks, trended increased protein levels of Nrf2 and nuclear Nrf2 (nNrf2), and the mRNA levels of the Nrf2 downstream genes (such as superoxide dismutase, SOD; and glutathione peroxidase, GPX), compared with control mice [46]. Early research from Xu, et al. [63] reported that the ob/ob mice had higher mRNA expression levels of heme oxygenase 1 (HO1), NAD(P)H: quinone oxidoreductase 1 (NQO1) in liver and HO1 in WAT, compared with WT mice. Moreover, this heightened adaptive antioxidant response was disrupted by Nrf2 ablation. These observations are consistent with research which has reported that feeding WT mice with HFD (60% kcal fat) for 180 days up-regulated the Nrf2 mRNA levels by approximately 50% in the liver, and by 14 times in the WAT [49].

Nevertheless, contradictory results emerged from earlier studies. The mRNA expression levels of Nrf2 and its targets glutathione transferase m6 (GSTm6) and NQO1 were found to be reduced upon 4-week HFD feeding in WT mice [64]. Data from He and colleagues demonstrated that the total and nuclear protein levels of Nrf2 in 18-week HFD (60% kcal fat)-fed mice were almost 50% lower in the skeletal muscle, compared with the ND-fed mice [48]. Moreover, Illesca and his team found the mRNA levels and DNA-binding activities of Nrf2 exhibited significant decreases in WAT with 12-week HFD (60% kcal fat) exposure [47]. These data indicated that functional Nrf2 is actually suppressed in response to HFD, whereas Li, et al. [65] observed that 12-week HFD (60% kcal fat) had no substantial impact on the mRNA levels of Nrf2 and its target genes NQO1 and HO1 in the liver of WT mice or hepatocyte-specific Nrf2 KO mice. What is more, in another study, HFD feeding significantly increased the nuclear but not total protein levels of Nrf2 in the liver. However, from the authors’ points of view, Nrf2 deficiency may have induced obesity and steatohepatitis in mice fed with HFD and the activation of Nrf2 could reduce these effects [56].

The inconsistent results concerning the effects of DIO on Nrf2 indicate that the differences in the diet composition (i.e., lard vs. soybean oil), fat content and the length of feeding period may be directly or indirectly correlated with the regulation of Nrf2. Table 2 summarizes the recent findings concerning the influence of HFD composition (fat content) and feeding duration on the activity of Nrf2. Obesity is strongly associated with oxidative stress in humans and rodent models [23]. When the body is exposed to obese danger, Nrf2, as a crucial regulator of antioxidant defense, is up-regulated to counter oxidative stress, suggesting an instinctive compensatory response from the organism. In response to obesity, a certain degree of Nrf2 activation exerts cellular protective effects via anti-oxidative stress, but it may seriously destroy the redox homeostasis and aggravate obesity if its activation exceeds the extent or duration. This may explain why some studies (see Table 1) showed that Nrf2 KO mice exhibited an improved obesity phenotype compared with WT mice. Early studies [66] supported a contention that the benefits of Nrf2 activation by knocking down Keap1 in acute toxicity were conclusive and that continuous Nrf2 activation beyond a certain threshold was rather disadvantageous to long-term survival. The above analysis showed that increased Nrf2 activity within a certain range and duration is protective in obesity. However, it is uncontrollable to fundamentally increase the activity of Nrf2 through genetic modification, so some studies have shown contradictory results (see Table 1). On the other hand, compared with genetic modification, the dosage and duration of treatment with Nrf2 activators can be artificially controlled. Therefore, from the review articles [39,67,68,69] published in recent years and Table 1 summarized in this paper, we found that Nrf2 activators, always as protectors, alleviate obesity and related metabolic diseases, showing decreased ROS production, inhibited lipid accumulation during adipogenesis, attenuated pro-inflammatory cytokines and improved glucose homeostasis. It may be possible in the future to develop effective and safe Nrf2 activators for the therapy of obesity.

## 3. Adipogenesis

The WAT of ob/ob mice exhibited elevated expression levels of fatty acid synthase (FAS) and peroxisome proliferator-activated receptor γ2 (PPARγ2), which involved in adipogenesis and lipogenesis, in contrast to WT mice, but this augmentation was diminished in global and adipocyte-specific Nrf2 KO ob/ob mice [63]. In an earlier study, the absence of Nrf2 resulted in attenuated expression levels of PPARγ, CCAAT enhancer-binding protein α (C/EBPα) and their downstream genes during adipocyte differentiation in employing 3T3-L1 cells, mouse embryonic fibroblasts (MEFs) or human subcutaneous preadipocytes. Simultaneously, enhanced Nrf2 activity by knocking down Keap1 reversed these changes [52]. HFD exposure also enhanced the mRNA and protein expression levels of PPARγ2 in liver, whereas the up-regulation of PPARγ2 was partially abrogated in Nrf2-deficient hepatocytes [65], suggesting that the regulatory effects of Nrf2 on PPARγ2 in hepatocytes may display a similar pattern as in adipocytes. What is more, the Nrf2 KO hepatocytes treated with palmitate which simulates PPARγ and consequent lipogenesis also showed lower mRNA levels of PPARγ and its downstream lipogenic genes, including the FAS, stearoyl-CoA desaturase 1 (SCD1) and fatty acid binding protein 4 (FABP4), than those in WT hepatocytes [65].

Similar to the aforementioned studies, Sun and colleagues discovered significantly decreased protein expression levels of acetyl-CoA carboxylase (ACC), sterol regulatory element binding protein 1 (SREBP1), diacylglycerol acyltransferase 1 (DGAT1), DGAT2, FAS and perilipin A in the WAT of Nrf2 KO mice, under both ND and HFD conditions [46]. However, it was also shown that the mRNA levels of DGAT2 in epididymal WAT (eWAT) were slightly increased instead of decreased by the loss of Nrf2 in adipocytes [43], and SREBP1 exhibited higher rather than lower mRNA expression levels in HFD-fed hepatocyte-specific Nrf2-deficient mice than the control counterparts [65]. Intriguingly, it was reported that sustaining Nrf2 activation reduced the triglycerides (TG) and free fatty acid (FFA) contents in Keap1-KD ob/ob mice, and weakened the mRNA expression of PPARγ, C/EBPα, SCD1, FABP4, ACC-1 and ACC-2. Moreover, after 36 days of HFD (60% kcal fat) feeding, Keap1-KD mice (non-ob/ob) had a lower body weight, with decreased food intake, compared to control mice [58]. The question emerges whether the improved obesity in Keap1-KD mice is the effect of Nrf2 activation, or an indirect result of decreased food intake caused by Keap1 gene manipulation. Wakabayashi, et al. [72] reported that Keap1-deficient mice (not KD, but KO) died from malnutrition resulting from severe hyperkeratosis in the esophagus and forestomach, indicating that the down-regulation or deletion of Keap1 may have affected the ingestive behavior of animals. In addition to the genetic manipulation of Keap1-Nrf2, recent studies using Nrf2 pharmacological regulators to investigate the role of Nrf2 on adipogenesis have also yielded conflicting results [73,74,75,76].

The in vitro experiments conducted by Xu and his team using MEFs suggested that the induction of Nrf2 mRNA expression in the early stages (0–12 h) of the differentiation from MEFs to adipocytes significantly declined in the later stage. What is more, after the early time point when MEFs differentiate into adipocytes, pharmacological Nrf2 activation by L-sulforaphane suppressed further differentiation and lipid accumulation [58]. L-sulforaphane was one of the first studied activators of Nrf2, belonging to the isothiocyanate group, present in broccoli. L-sulforaphane-rich broccoli sprout powder significantly improved serum insulin concentration, glucose-to-insulin ratio and insulin resistance in type 2 diabetic patients [77]. These findings are similar to those from Chartoumpekis, et al. [78]. These data indicated that Nrf2 was essential for adipocyte differentiation, which is a result consistent with most studies demonstrating that Nrf2 deficiency restrains fat accumulation in WAT. However, for differentiated or differentiating adipocytes, Nrf2 may play an antagonistic role in lipid accumulation, which partially explains why various research has shown that Nrf2 activation ameliorates obesity. In another study, treatment with punicalagin, which improves obesity via activating Nrf2, substantially inhibited lipid accumulation in the early stages (0–2 days) of 3T3-L1 cell differentiation. However, these effects were dramatically attenuated after that time point [59]. An intriguing finding was also observed in an in vivo experiment. Adipocyte-specific Nrf2 KO (ANKO) mice exhibited transiently delayed BW growth from week 5 to week 11 of HFD feeding; nonetheless, after 14 weeks, ANKO mice showed comparable BW and body fat content with control mice [50]. In summary, the effects of time specificity of Nrf2 to fat remains unclear and requires further study.

## 4. Inflammation

Since the discovery [79] that the pro-inflammatory cytokine tumor necrosis factor α (TNFα) induced by obesity was able to promote IR, numerous studies have consistently shown increased inflammation responses in WAT of obese humans and animals [16,24,80,81,82,83,84]. While contributions of oxidative stress to inflammation have been recognized, contradictory reports exist regarding the function of Nrf2 in obesity-induced inflammation. Xue and his team employed global or adipocyte-specific Nrf2-knockout (KO) mice on leptin-deficient (ob/ob) background to investigate the roles of Nrf2 in obesity and its associated disorders. They found augmented inflammatory responses determined by increased expression levels of pro-inflammatory cytokines, including TNFα and interleukin 1β (IL-1β), in WAT of ob/ob mice, compared with non-ob/ob littermates [63]. This is consistent with the view that chronic subclinical inflammation might be involved in the pathogenesis of obesity [16,84,85,86,87,88]. However, either systemic or adipocyte-specific ablation of Nrf2 in ob/ob mice can unexpectedly improve inflammation, as Nrf2 has been shown to be able to suppress inflammation [38,89,90]. In a more recent study, the deficiency of Nrf2 in hepatocytes of mice attenuated inflammation induced by feeding HFD (60% kcal fat) for 12 weeks, as suggested by the suppressed phosphorylation of nuclear factor kappa-B (NF-κB), the mRNA degrees of TNFα and C–C motif chemokine ligand 2 (CCL2) and minimized macrophage infiltration [65]. These findings are also in agreement with a previous study conducted by More and colleagues. They observed that constitutive Nrf2 activation by Keap1 konckdown (KD) exaggerated inflammation through increasing the mRNA expression of TNFα in both WAT and liver of C57BL/6 mice fed with long-term (24 weeks) HFD (60% kcal fat) [51].

Why is it that Nrf2 is widely considered as an anti-inflammatory factor [38,89,91], and yet so many studies have located that the absence of Nrf2 can ameliorate inflammation in metabolic syndrome? Nrf2 has been shown to promote adipogenesis by up-regulating the expression of PPARγ in WAT [52] and liver [65]. Thus, continuous activation of Nrf2 under stress conditions may increase lipid accumulation and lead to lipid peroxidation, which in turn causes tissue injury and inflammation. The pro-inflammation might not be the direct effect of Nrf2, but its secondary effect mediated by promoting adipogenesis. On the other hand, Nrf2 modulates inflammatory responses in a cell type-dependent manner, which may also partially contribute to the contradictory effects of Nrf2 on inflammation. Nrf2 acts as a transcription factor to activate the expression levels of IL-6 in hepatocytes [92], which however block the transcription of IL-6 and IL-1β in macrophages [89] by inhibiting the recruitment of RNA polymerase Ⅱ. A study from Meher and colleagues found that the mRNA expression levels of monocyte chemoattractant protein 1 (MCP-1) and IL-1β in macrophages were up-regulated when co-culturing with adipocytes, but this up-regulation was significantly suppressed in Nrf2 KO macrophages. However, co-culturing with Nrf2-deficient macrophages induced the mRNA expression of IL-6 and MCP-1 in adipocytes [45]. This study implied that Nrf2 plays an important role in promoting the immunological function of macrophages (participating in inflammation), but the activity of Nrf2 in macrophages did not affect the response of adipocytes to inflammatory signals, and adipose tissue inflammation may have been mainly regulated by Nrf2 in adipocytes rather than macrophages (Figure 2). In addition to the specific roles of Nrf2 in different cell types, the crosstalk between cells may also have a great influence on the effects of Nrf2 on inflammation.

## 5. Insulin Resistance

Oxidative stress is a potent inducer of insulin resistance which is a pathologic condition during the development of obesity [15,60,93]. A few studies have confirmed Nrf2 is related to insulin signaling and insulin resistance due to its crucial cytoprotective, ROS scavenging, and anti-inflammatory roles [39,83,93,94,95]. However, the precise underlying mechanism by which Nrf2 exerts its effects on insulin resistance is still obscure. Uruno, et al. [96] over-expressd Nrf2 globally and specifically in the skeletal muscle (SkM) or liver by Keap1 gene hypomorphic knockdown and SkM- or liver-specific Keap1 knockout. Their results showed that either systemic or SkM-specific KO/KD of Keap1 in db/db mice decreased blood glucose and elevate insulin sensitivity. However, these changes were not seen in liver-specific Keap1-KO db/db mice. Moreover, they employed Keap1-KD mice (non-db/db) fed with HFD (62.2% kcal fat) for 8 weeks, and Nrf2 exhibited the same ameliorated IR effects. Subsequently, it was noted that Nrf2 induction in SkM reduced glycogen content, resulting in improved glucose tolerance [97]. These observations are similar to a report which found that Nrf2 ablation in HFD-treated mice can lead to hepatic IR by activation of the NF-κB signaling pathway [56]. Chartoumpekis, et al. [98] also found that increased Nrf2 activity in Keap1-KD lipodystrophic mice alleviated impaired metabolism and IR via inhibition of lipogenic genes in the liver.

Inconsistent with the above studies, data from More and colleagues suggested that after 24 weeks of HFD (60% kcal fat) feeding, Keap1-KD mice had higher blood glucose with aggravated IR, compared to control mice [51]. The same study showed that Nrf2 activation induced by knocking down Keap1 suppressed insulin signaling and deteriorated IR in ob/ob mice [58]. These findings are similar to those by Meakin, et al. [99] and Chartoumpekis, et al. [49]. They reported that Nrf2 KO mice displayed improved insulin sensitivity when exposed to HFD.

Fibroblast growth factor 21 (FGF21), which is a stress-inducible peptide hormone that regulates energy balance and glucose and lipid homeostasis, has been proven to alleviate hyperglycemia, insulin resistance, dyslipidemia and fatty liver [100,101,102,103]. HFD-fed Nrf2-null mice had higher circulating FGF21 levels and increased FGF21 mRNA levels in WAT and liver compared with WT mice. Furthermore, the over-expression of Nrf2 in ST-2 cells, which is a cell line derived from murine bone marrow stroma that expresses functional FGF21 and Nrf2, led to a decline in FGF21 mRNA expression. As a result, the improved obesity and IR phenotype of Nrf2 KO mice may be partly attributed to the up-regulation of FGF21 induced by Nrf2 ablation [49]. However, there have been conflicting results in a more recent study showing that FGF21 is obviously repressed in Nrf2-deficient 3T3-L1 cells, using small interfering RNA (siRNA) [104]. More interestingly, FGF21 has been reported to inhibit inflammation by activating Nrf2 and suppressing the NF-κB signaling pathway [105].

To clarify which tissue contributes to the changes in IR brought by Nrf2, Chartoumpekis and colleagues generated mice with conditional knockout of Nrf2 in hepatocytes (HeNKO) or adipocytes (ANKO). After 170 days of HFD (60% kcal fat) feeding, HeNKO mice and ANKO mice revealed comparable increasing levels in body weight and body fat mass. However, ANKO mice exhibited higher fasting glucose, cholesterol levels and disruptive glucose tolerance compared with control group; deletion of Nrf2 in hepatocytes inversely led to an ameliorated metabolic profile (improved insulin sensitivity and glucose tolerance) without any difference in hepatic steatosis. It is interesting that the FGF21 mRNA levels showed no significant difference in both WAT of ANKO mice and liver of HeNKO mice, and FGF21 plasma levels werealso similar among all genotypes [43].

In summary, the inconsistent results on the role of Nrf2 in IR from the above studies indicate that the difference in genetic manipulation of Nrf2 (Nrf2 Knockout or Keap1 Knockdown), length of HFD feeding period, duration of Nrf2 activation and cell specificity of Nrf2’s function, may lead to different effects on IR. The study of Loh, et al. [106] may explain the effects of Nrf2 activation duration on insulin signaling. Although the excess ROS is strongly related to the pathophysiology of many diseases, including diabetes and obesity, actually physiological levels of ROS may be necessary for normal intracellular signaling. Loh and colleagues demonstrated subtle increases in physiological ROS production enhance insulin signaling in vivo. One potential explanation is as follows. Short-term Nrf2 activation improves IR through scavenging the excessive ROS, however, continuous Nrf2 activation impair insulin signaling by excessively removing ROS. More future studies focusing on the regulation of ROS homeostasis by Nrf2 are needed. Any coin has two sides and even the roles of inflammation in IR have inspired different views in recent years [107,108,109,110].

## 6. Conclusions

In this review, we discussed Nrf2’s subtle expression patterns in response to obesity and the complex roles of Nrf2 on adipogenesis, inflammation and insulin resistance. In obesity, chronic oxidative stress is considered to be the important factor that leads to the development of pathologies such as lipid disorders, insulin resistance and diabetes. Nrf2 is a key regulator of the cellular response to oxidative stress by controlling the expression of antioxidant and detoxification enzymes to eliminate excess ROS. An increasing number of studies have provided casual evidence that Nrf2 is a pivotal target of obesity and its related metabolic disorders. Of note, previous reports have had many contradictions about the roles of Nrf2 in obesity and its related disorders. As indicated by the above analysis, further study on the spatio-temporal specificity of Nrf2 regulating cell function, the final effects of Nrf2 crosstalk between different cell types on the overall function of the body, the biological effects of different gene manipulation of Nrf2, and the regulation of ROS homeostasis by the activation degree and duration of Nrf2 will undoubtedly help clarify the mentioned inconsistence and lead to a greater understanding of the essential roles of Nrf2 in the modulation of metabolic homeostasis, and would provide novel insights for potential therapeutic approaches in obesity and its comorbidities.

## Figures and Tables

**Figure 1 antioxidants-11-02067-f001:**
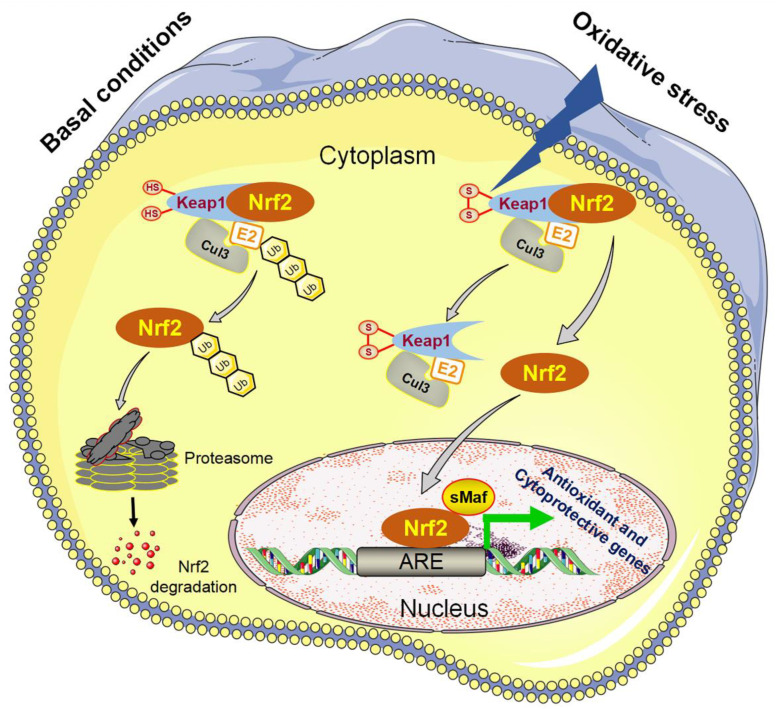
Schematic diagram of Keap1-Nrf2-ARE signaling pathway. Under basal non-stressed conditions, Keap1 binds to cullin 3 (Cul3)-based E3 ubiquitin ligases, forming Keap1-Cul3-Nrf2 complex, leading to Nrf2 ubiquitination and proteasomal degradation. In response to oxidative stress, some cysteine residues of Keap1 are oxidized, resulting in a conformation change in Keap1 and disruption of the Keap1–Cul3 interaction. Nrf2 then dissociates from Keap1 and translocates to the nucleus, where it forms a heterodimer with small Maf proteins (sMaf), binds to ARE and stimulates the transcription of antioxidant and cytoprotective genes.

**Figure 2 antioxidants-11-02067-f002:**
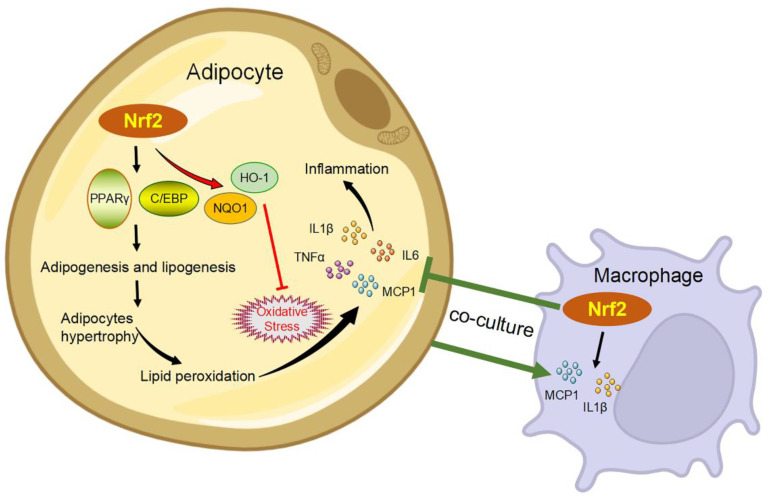
The subtle role of Nrf2 in obesity-related inflammation. Nrf2 promotes adipogenesis and lipogenesis by up-regulating PPARγ and C/EBP in adipocytes. Thus, continuous activation of Nrf2 under obesity-induced stress conditions may uncontrollably increase lipid accumulation and lead to lipid peroxidation, which in turn causes tissue injury and inflammation. At the same time, Nrf2, as a double-edged sword, counteracts oxidative stress by enhancing the expression of NQO1 and HO-1 and subsequently alleviates inflammation. The mRNA levels of MCP-1 and IL-1β in macrophages are up-regulated when co-culturing with adipocytes, but this up-regulation is significantly suppressed in Nrf2 KO macrophages. However, co-culturing with Nrf2-deficient macrophages induces the mRNA expression of IL-6 and MCP-1 in adipocytes.

**Table 1 antioxidants-11-02067-t001:** Different roles of Nrf2 signaling in animal models of diet-induced obesity (DIO).

Mice	Age	Body Weight (BW)	Diet	Treatment	Impaired Glucose Homeostasis	Obesity	References
C57; WT	5 weeks	17–20 g	45% fat, 28 weeks	HFD plus oltipraz (0.75 g/kg diet)	↓	↓	[40]
C57; WT	6–7 weeks	16–18 g	60% fat, 21 days or 95 days	Oral gavage with CDDO-Im (30 μmol/kg BW) 3 times per week throughout the feeding period	Unreported	↓	[41]
C57; WT	7 weeks	20–25 g	60% fat, 14 weeks	HFD containing 0.3% glucoraphanin	↓	↓	[42]
C57; Nrf2-HeNKO, Nrf2-ANKO	8 weeks	25–30 g	60% fat, 170 days	Nrf2 KO in hepatocytes (HeNKO) or adipocytes (ANKO)	Nrf2-HeNKO↓, Nrf2-ANKO↑	Nrf2-HeNKO↓, Nrf2-ANKO↑	[43]
C57; Nrf2-KO, Keap1-KD	12 weeks	25–30 g	39.7% fat, 12 weeks	Nrf2-KO, Keap1-KD	Nrf2-KO↓, Keap1-KD↑	Unaltered	[44]
C57; Nrf2-KO, Nrf2-mKO	8–12 weeks	22–27 g	60% fat, 10 weeks	Nrf2-KO, myeloid Nrf2-KO (Nrf2-mKO)	Nrf2-KO↓, Nrf2-mKO Unaltered	Nrf2-KO↓, Nrf2-mKO Unaltered	[45]
C57; Nrf2-KO	4 weeks	~20 g	41% fat, 16 weeks	Nrf2-KO	Unreported	↓	[46]
C57; WT	Unreported	12–14 g	60% fat, 12 weeks	Oral gavage with 5 mg Hydroxytyrosol/kg BW/day	Improve the WAT dysfunction		[47]
C57; WT	8 weeks	15–18 g	60% fat, 16 weeks + 15 days	Curcumin was given daily by oral gavage at the dose of 50 mg/kg BW for 15 days	↓	Unaltered	[48]
C57; Nrf2-KO	9–10 weeks	25–28 g	60% fat, 180 days	Nrf2-KO	↓	↓	[49]
C57; WT	7 weeks	22–25 g	60% fat, 12 weeks	Treated with two doses of oral administration of sesamol (100 mg/kg/day or 200 mg/kg/day) for 12 weeks	↓	↓	[50]
C57; Keap1-KD	3 weeks (start HFD)	16–24 g (6 weeks, start measuring BW)	60% fat, 24 weeks	Keap1-KD	↑	↑	[51]
C57; Nrf2-KO	4 weeks	~20 g	41% fat, 24 weeks	Nrf2-KO	Unreported	↓	[52]
C57; Nrf2-KO	12–16 weeks	~20 g	60% fat, 6 weeks	Nrf2-KO	↓	↓	[53]
C57; Nrf2-ANKO	6 weeks	~30 g	60% fat, 14 weeks	Nrf2 KO in adipocytes (ANKO)	↓	↓ in 5–11 weeks and comparable with WT mice after 14 weeks	[54]
C57; Keap1-KD	8 weeks	15–20 g	60% fat, 90 days	Keap1-KD	↓	↓	[55]
ICR; Nrf2-KO	6–8 weeks	22–24 g	10% lard, 2% cholesterol, 0.5% bile salt and 87.5% base forage (Not reported total fat), 8 weeks	Nrf2-KO	↑	↑	[56]
C57; WT	5 weeks	22–24 g	45% fat, 17 weeks	EGCG at 25 or 75 mg/kg were administered via i. p. three times a week over a period of seventeen weeks	↓	↓	[57]
C57; Keap1-KD	9 weeks	22–26 g	60% fat, 36 days	Keap1-KD	Unreported	↓	[58]
C57; WT	6 weeks	20–25 g	60% fat, 14 weeks	PCG at 10 or 100 mg/kg was administered by oral gavage five times a week	↓	Unreported (Inflammation↓)	[59]

Note: Oltipraz, CDDO-Im, glucoraphanin, hydroxytyrosol, curcumin, sesamol, EGCG, and PCG are considered to be the specific activators of Nrf2 or exert their biological functions by activating Nrf2. KO: knockout; KD: knockdown; HFD: high-fat diet. ↓ indicates decreased/ameliorated; ↑ indicates increased/deteriorated.

**Table 2 antioxidants-11-02067-t002:** Influence of HFD composition (fat content) and feeding duration on Nrf2 activity.

Diet Source and Identifier	Fat Calories	Fat Composition (*w*/*w*)	Feeding Duration	Effects on Nrf2	References
Bio-Serv, F3282	60%	Lard (36.0%)	10 weeks	Decreased Nrf2 mRNA levels in WAT and liver	[45]
Research Diets, D12492	60%	Lard (31.7%), Soybean Oil (3.2%)	24 weeks	Increased Nrf2 mRNA levels in WAT	[51]
Research Diets, Unreported	60%	Unreported	26 weeks	Increased Nrf2 mRNA levels in WAT and liver	[49]
Research Diets, D12492	60%	Lard (31.7%), Soybean Oil (3.2%)	12 weeks	Decreased Nrf2 DNA binding activity and mRNA levels in eWAT	[47]
Harlan Teklad, TD88137	42%	Anhydrous Milkfat (21.0%)	12 weeks	Increased Nrf2 mRNA levels in liver	[70]
Research Diets, D12042201	41%	Anhydrous Butter (20.0%)	16 weeks	Increased total and nuclear protein levels of Nrf2 in WAT	[46]
Guangdong Animal Center, Unreported	60%	Unreported	18 weeks	Decreased total and nuclear protein levels of Nrf2 in skeletal muscle	[48]
Harlan Teklad, TD10885	45%	Anhydrous Milkfat (21.0%), Soybean Oil (2.0%)	28 weeks	Decreased nuclear but not total protein levels of Nrf2 in WAT	[40]
Unreported, Unreported	36%	Soybean Oil (18.0%)	12 weeks	Increased Nrf2 mRNA levels in liver	[71]
Bio-Serv, F5194	Unreported	Lard (35%), cholesterol (0.15%)	4 weeks	Decreased mRNA levels of Nrf2 and downstream NQO1 and GSTm6 in liver	[64]
Research Diets, D12492	60%	Lard (31.7%), Soybean Oil (3.2%)	12 weeks	No significant changes in mRNA or protein levels of Nrf2	[65]

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
