# Peer review of "The Nrf2 in Obesity: A Friend or Foe?"

_antioxidants, 2022, doi:10.3390/antiox11102067_

Round 1

Reviewer 1 Report

This is a very interesting and well written review highlighting the multifaceted role of NRF2/KEAP1 signaling pathway in obesity. Only two minor points could be improved. In particular:

Introduction: It deserves to be highlighted that, although NRF2 activation is an important antioxidant modulator in normal cells, this effect is also one of the main factor involved in chemo- and radio-resistance in cancer cells since many studies are focused to reduce Nrf2 activation in chemoresistant and radioresistant cancer cells  (see PMID: 35901941, 33287295, 35368867

Lines 182-186: The authors need to underline that L-sulforaphane an isothiocyanate present in broccoli, was one of the first studied activators of Nrf2. In type 2 diabetic patients, L-sulforaphane-rich broccoli sprout powder caused a significant improvement in serum insulin concentration, glucose-to-insulin ratio, and insulin resistance (PMID: 33123312).

Author Response

Dear reviewer,

Thank you very much for your valuable and professional opinions, these comments are very helpful for revising and improving our paper. We have studied comments carefully and have made correction which we hope meet with approval. Any revisions made to the manuscript have been marked up in red using the “Track Changes” function. Furthermore, we would like to show the details as follows:

Point 1: Introduction: It deserves to be highlighted that, although NRF2 activation is an important antioxidant modulator in normal cells, this effect is also one of the main factors involved in chemo- and radio-resistance in cancer cells since many studies are focused to reduce Nrf2 activation in chemoresistant and radioresistant cancer cells (see PMID: 35901941, 33287295, 35368867).

Response 1: We are very sorry that we ignored the critical role of Nrf2 in cancer cells. We once read an article entitled “NRF2 activation by antioxidant antidiabetic agents accelerates tumor metastasis” which indicated that antioxidants that activate Nrf2 signaling may need to be administered with caution in cancer patients, such as diabetic patients with cancer (PMID: 27075625). More importantly, we have carefully read the three articles you shared with us. It is necessary to briefly introduce the effect of Nrf2 on cancer cells. So, we have changed “Nevertheless, accumulating evidence has demonstrated that the role of Nrf2 may overstep oxidative stress protection.” to “Nrf2 is like a coin with two sides. Although Nrf2 activation is an important antioxidant modulator in normal cells, this effect is also one of the main factors involved in the enhanced resistance to chemotherapy and radiotherapy in cancer cells since many studies are focused to reduce Nrf2 activation in chemoresistant and radioresistant cancer cells [34-36]. More remarkably, Nrf2 activation by antidiabetic agents has been reported to accelerate tumor metastasis [37].” (Page 2, line 67-73)

Point 2: Lines 182-186: The authors need to underline that L-sulforaphane an isothiocyanate present in broccoli, was one of the first studied activators of Nrf2. In type 2 diabetic patients, L-sulforaphane-rich broccoli sprout powder caused a significant improvement in serum insulin concentration, glucose-to-insulin ratio, and insulin resistance (PMID: 33123312).

Response 2: As you suggested, on page 8, line 203-206, a sentence has been added: “L-sulforaphane is one of the first studied activators of Nrf2, belonging to the isothiocyanate group, present in broccoli. L-sulforaphane-rich broccoli sprout powder significantly improved serum insulin concentration, glucose-to-insulin ratio, and insulin resistance in type 2 diabetic patients [77].”

We would like to thank you again for taking the time to review our manuscript. Looking forward to hearing from you.

Yours sincerely,

Yudong Xia

E-mail: yudongxia@stu.xjmu.edu.cn

Reviewer 2 Report

Yudong Xia et al made an attempt to illuminate the complicated interplay among Nrf2, oxidative stress, lipid metabolism, insulin signalling and chronic inflammation in obesity. Manuscript is well written and the question posed by the authors is well defined. The manuscript adheres to the relevant standards for reporting.  However, few suggestions to strengthen the manuscript.

The authors should briefly describe the Nrf2 pathways/function and its activation process.

It would be better to describe about on the role of Nrf2 activators on the interplay between Nrf2, oxidative stress, lipid metabolism, insulin signalling.

The authors are requested to go through these papers to strengthen the manuscript. PMID: 34455076, 32971975, 25540093, 23319124

Author Response

Dear reviewer,

Thank you very much for your valuable and professional advice, these comments extremely help to improve the academic quality of our paper. Based on your suggestions, we have carefully made modifications in the revised manuscript. Any revisions made to the manuscript have been marked up in red using the “Track Changes” function. The point-to-point response to your comments are as follows:

Point 1: The authors should briefly describe the Nrf2 pathways/function and its activation process.

Response 1: Thanks for your comments, the Nrf2 pathways/function and its activation process have been already described in Introduction (Page 2, line 55-66) and Figure 1 (Page 5).

Point 2: It would be better to describe about on the role of Nrf2 activators on the interplay between Nrf2, oxidative stress, lipid metabolism, insulin signaling.

Response 2: We appreciate the insightful suggestion, which has been very constructive in refining this review. After studying the four papers (PMID: 34455076, 32971975, 25540093, 23319124) you shared with us carefully, we can find that Nrf2 activators, always as protectors, alleviate obesity and related metabolic diseases, showing decreased ROS production, inhibited lipid accumulation during adipogenesis, attenuated pro-inflammatory cytokines and improved glucose homeostasis. Such consistent results are rarely seen in gene-modified animals. This phenomenon may be explained with one conjecture that increased Nrf2 activity within a certain range and duration is protective in obesity. However, it is uncontrollable to fundamentally increase the activity of Nrf2 through genetic modification, so some studies have shown contradictory results (see Table 1 on page 2-4). On the other hand, compared with genetic modification, the dosage and duration of treatment with Nrf2 activators can be artificially controlled. It may be possible in the future to develop effective and safe Nrf2 activators for the therapy of obesity. Related revisions to the manuscript can be seen on page 6, line 144-154.

We would like to express our great appreciation to you for comments and suggestions on our paper. Looking forward to hearing from you.

Yours sincerely,

Yudong Xia

E-mail: yudongxia@stu.xjmu.edu.cn
